# Attitudes towards a programme of risk assessment and stratified management for ovarian cancer: a focus group study of UK South Asians' perspectives

Katie E J Hann,[1,2] Nasreen Ali,[3] Sue Gessler,[1] Lindsay Sarah Macduff Fraser,[1] Lucy Side,[1,4] Jo Waller,[5] Saskia C Sanderson,[5,6] Anne Lanceley,[1] for the PROMISE study team

For numbered affiliations see end of article.

**Correspondence to**
Dr Anne Lanceley;
a.lanceley@ucl.ac.uk

## ABSTRACT

**Objective** Population-based risk assessment, using genetic testing and the provision of appropriate risk management, could lead to prevention, early detection and improved clinical management of ovarian cancer (OC). Previous research with mostly white British participants found positive attitudes towards such a programme. The current study aimed to explore the attitudes of South Asian (SA) women and men in the UK with the aim of identifying how best to implement such a programme to minimise distress and maximise uptake.

**Design** Semistructured qualitative focus group discussions.

**Setting** Community centres across North London and Luton.

**Participants** 49 women and 13 men who identified as SA (Indian, Pakistani or Bangladeshi), which constitutes the largest non-European ethnic minority group in the UK.

**Methods** Seven community-based focus groups were held. Group discussions were transcribed verbatim, coded and analysed thematically.

**Results** Awareness and knowledge of OC symptoms and specific risk factors was low. The programme was acceptable to most participants and attitudes to it were generally positive. Participants' main concerns related to receiving a high-risk result following the genetic test. Younger women may be more cautious of genetic testing, screening or risk-reducing surgery due to the importance of marriage and childbearing in their SA cultures.

**Conclusions** A crucial first step to enable implementation of population-based genetic risk assessment and management in OC is to raise awareness of OC within SA communities. It will be important to engage with the SA community early on in programme implementation to address their specific concerns and to ensure culturally tailored decision support.

## INTRODUCTION

Ovarian cancer (OC) is the sixth most common cancer among UK women.[1] Due to the non-specific symptoms associated with

### Strengths and limitations of this study

► This is the first study to explore the attitudes of a UK ethnic minority group towards population-based risk assessment and stratified management for ovarian cancer.
► The study explored the attitudes of both women and men.
► Opinions solicited during the focus groups were directly related to information provided about population-based risk assessment and stratified management for ovarian cancer and this may have limited responses.
► Two female researchers facilitated all the focus groups including those with men, this may have influenced the findings.

this cancer, diagnosis is usually at a late stage when prognosis is poor.[2]

Earlier detection of OC could help to save lives and this has fuelled voluntary sector demands for research to investigate approaches for prevention and earlier diagnosis.[3] A definitive ongoing trial investigating screening for OC in postmenopausal women has shown this to be sensitive and feasible,[4 5] but to date without a significant mortality benefit.[5 6] Nevertheless, a stage shift at diagnosis has been evidenced and for high-risk women who are not ready to have risk-reducing surgery,[5] screening could be an interim option.

Mutations of *BRCA1* and *BRCA2* genes considerably increase an individual's risk of OC;[7] combined with non-genetic information (eg, family history of cancer, age and lifestyle factors), this genetic information can be used to estimate a woman's risk. Following risk assessment, risk-stratified management could

benefit patients by identifying those at high risk and in most need of management, while avoiding overinvestigation of those at lowest risk.[8 9] A current programme of research, Predicting Risk of Ovarian Malignancies, Improved Screening and Early Detection (PROMISE, https://eveappeal.org.uk/our-research/our-research-programmes/promise-2016/), involves a feasibility trial to investigate whether stratified OC risk management is acceptable to women in the UK general population. In this programme, women will be provided with an estimate of their OC risk and stratified as low, intermediate or high risk. Those with the lowest risk will be provided with information on OC. Screening or surgery to remove the ovaries will be offered to those at intermediate and high risk. In the future, equivalent programmes could be rolled out for other cancers.[10]

Positive attitudes to the PROMISE programme were reported among women in the general[11] and in high OC risk populations,[12] but study samples did not reflect the diverse UK population. A key concern for any public health programme is its inclusivity, yet little is known about UK ethnic minority peoples' awareness and attitudes towards genetic testing for cancer risk.[13] The few studies that have explored delayed use of genetic services among UK minority groups identify low awareness of their availability, language barriers and unwillingness to discuss cancer due to stigma and fear as contributing factors.[14 15]

This study aimed to explore South Asian (SA) women's and men's attitudes towards the PROMISE programme and the idea of population-based genetic testing and risk-stratified management of OC, and to identify factors which may influence participation. The SA community is the largest non-European ethnic minority group in the UK, over 5% of the population in England and Wales identify as SA (Pakistani, Bangladeshi and Indian).[16] The attitudes of both men and women were explored because it is known that healthcare decisions may be influenced by family members including husbands and/or fathers.[17 18]

## METHODS
### Methodological approach
The study took a constructionist perspective in which meaning and experience are considered to be socially produced and reproduced rather than as immutable individual characteristics.[19] It used a qualitative research design of focus groups to explore existing knowledge of OC and views of the novel population-based risk management intervention. Focus groups are well suited to exploration of public health topics and are a good way of identifying community norms and cultural values.[20] Structured discussion within the groups provided an opportunity for participants to question each other and reflect on and challenge one another's views. Thematic analysis of these data was undertaken.

### Patient/public involvement
Patients and the public were involved as project steering group members in the design of the overall PROMISE programme and the health behaviour workstream within which this study was delivered. Members of the SA community also contributed by pilot testing our presentation materials for the focus groups.

### Setting
Participants were recruited from the North London Boroughs of Brent, Newham and Tower Hamlets and Luton, areas which have large SA settler communities. Groups were conducted in suitable local community venues between November 2016 and April 2017.

### Participants
Purposive sampling was used to include only individuals ≥18 years old, who self-identified as being of SA ethnicity (Indian, Pakistani, Bangladeshi) and to include a wide spread of ages. At least some conversational English language was needed to take part. Men were included in the study as they may play a role in supporting and advising female family members' healthcare decisions. Women were excluded if they had (1) a diagnosis of OC and/or (2) previously had genetic testing to find out about personal cancer risk.

SA women and men were introduced to the study by local community centre staff (n=53), and through poster and leaflet advertisement at community centres and by a local women's health organisation (n=9). The few eligible individuals who contacted the research team directly by phone or email were sent the study information and had an opportunity to ask questions. We aimed to obtain a broad range of views and continued to recruit until we achieved data saturation,[21] when no new views were being expressed. Of those who agreed to participate, two withdrew due to sickness. Participants received a £20 gift voucher and travel costs.

### Data collection
Seven focus group discussions were held at community centres: five with women (n=12, n=8, n=9, n=11, n=9) and two with men (n=7, n=6). Each discussion lasted approximately 75 min. Groups were facilitated by NA, a SA multilingual senior qualitative researcher and KEJH, a research assistant with a master's level qualification, acting as moderator and note-taker alternatively. A semistructured discussion guide developed from previous work[11] and the literature and which comprised open-ended, none-directive questions, was used (see online supplementary additional files 1 and 2). These aimed to facilitate discussion and elicit participant views.

At the start of each focus group, KEJH and NA introduced themselves briefly (name, job, associated university), stated the study's purpose and confirmed what participation involved. Intragroup confidentiality, audio recording and study report confidentiality were highlighted with an opportunity to ask questions. Each participant completed a demographic questionnaire.

To open the discussion participants were invited to share their current awareness and knowledge of OC. Essential information concerning OC, including the increased risk among those with *BRCA1/2* gene mutations, the possibility to test for these and the PROMISE programme's proposal to offer OC risk-stratified management (see online supplementary additional file 3), was then given in a short slide presentation and handout. It was also explained that OC risk information from genetic testing could be less accurate for women of SA ethnicity as most research has been carried out with women of European descent.[22] The presentation text was designed to be understood by participants irrespective of educational attainment and was pilot tested with SA women for comprehension.

Two groups were conducted solely in English, and five in multiple languages including English, Urdu, Hindi, Punjabi, Pahari and Bengali. In two groups involving Bangladeshi women, those fluent in English assisted their peers so that everyone understood the language used. In one group which included Bangladeshi men, a woman acted as a translator to help a few participants take part in the discussion.

## Analysis
Group discussions were audio recorded, translated into English if necessary, transcribed verbatim by a professional multilingual transcription service and checked against the recordings for accuracy by KEJH and NA. The data were analysed thematically[23] using QSR International's NVivo V.10 Software (2012). KEJH read and reread the transcripts and generated initial codes. AL and NA also read the transcripts to identify any divergent cases, and initial codes were refined after discussion. Themes were identified deductively, guided by the discussion topics and inductively, as they emerged from the data. KEJH analysed all seven transcripts and an independent researcher (SG) coded two transcripts. KEJH and SG met to confirm any divergent cases and discuss any disagreements in coding until a consensus was reached. This paper follows the Consolidated Criteria for Reporting Qualitative Studies.[24]

## FINDINGS
A total of 49 women and 13 men took part. Demographic characteristics of participants are presented in table 1. Five themes were identified: participants' awareness and knowledge of OC and genetic risk; attitudes towards genetic testing and finding out about OC risk; attitudes towards risk-stratified management; family, culture and religion; and accessing services.

### Awareness and knowledge of OC and genetic risk
The term ovary/ovaries was not familiar to many participants. In all groups there was some confusion over the ovaries, what they are and where they are located in the body. Some participants had difficulty distinguishing

| Table 1 Sample demographics (n=62) | |
|---|---|
| | n (%) |
| **Gender** | |
| Female | 49 (79.0) |
| Male | 13 (21.0) |
| **Age** | |
| Mean years (range) | 50.5 (22–82) |
| **Ethnic group** | |
| Bangladeshi | 31 (50.0) |
| Indian | 14 (22.6) |
| Pakistani | 15 (24.2) |
| Other, Kashmiri | 2 (3.2) |
| **Approx. years lived in the UK** | |
| Mean (range) | 28.0 (2–49) |
| **First language*** | |
| English | 8 (12.9) |
| Bengali/Bangla | 32 (51.6) |
| Gujarati | 3 (4.8) |
| Hindi | 5 (8.1) |
| Pahari | 5 (8.1) |
| Punjabi | 4 (6.5) |
| Sylheti | 1 (1.6) |
| Urdu | 15 (24.2) |
| Missing | 1 (1.6) |
| **Religion** | |
| Hindu | 7 (11.3) |
| Muslim | 52 (83.9) |
| Sikh | 3 (4.8) |
| **Marital status** | |
| Married/living with partner | 44 (71.0) |
| Single/separated/divorced/widowed | 17 (27.4) |
| Missing | 1 (1.6) |
| **Employment** | |
| Full-time employment | 4 (6.5) |
| Part-time employment | 8 (12.9) |
| Home maker | 14 (22.6) |
| Retired | 14 (22.6) |
| Disabled/too ill to work/full-time carer | 3 (4.8) |
| Unemployed | 19 (30.6) |
| **Education** | |
| Degree or higher | 12 (19.4) |
| Qualification below degree level | 19 (30.64) |
| Still studying | 1 (1.8) |
| Other | 11 (17.7) |
| No formal qualifications | 19 (30.6) |

Continued

**Table 1** Continued

|  | n (%) |
|---|---|
| Attended screening | |
| Among female participants (breast or cervical screening or FOBT) | 39 (79.6) |
| Among male participants (FOBT) | 2 (15.4) |
| Cancer within social network | |
| Yes | 29 (46.8) |
| No/not sure/prefer not to say | 33 (53.2) |
| Personal cancer diagnosis | |
| Yes | 3 (4.8) |
| No/not sure/prefer not to say | 59 (95.2) |

*Some participants had more than one first language.
FOBT - Faecal Occult Blood Test

between the ovaries and the womb and this was reflected in some Urdu speakers using the word 'bacha daani' (womb) and 'undah daani' (ovaries) interchangeably. Most participants were aware of the UK's common cancers mentioning breast, prostate and lung, as well as cervical cancer, and correctly named some of the main risk factors for these. However, the majority had not come across OC and were unaware of the risk factors and main symptoms of the disease.

*Ovarian cancer…no one's heard of it.* FG4, woman, Luton.

*…I'm quite aware, I do pick up leaflets and read but I don't think I've come across ovarian cancer, not on TV, not on any sort of media, not on the train, nothing.* FG3, woman, London.

A few women incorrectly believed that use of hormone replacement therapy or the contraceptive pill would increase a woman's risk of OC. Older age was infrequently reported as a risk factor and few participants spontaneously spoke about family history or genetic risk. A minority of participants who demonstrated awareness of OC explained that this was due to either having researched the topic online prior to the group discussion (n=1) or from experience of a relative with OC (n=1).

Likely due to the lack of awareness, the women had not considered their risk of OC. Some indicated that they did not generally think about their personal risk of cancer. When asked, most acknowledged that they would have some risk of OC, although whether they perceived this to be the same, lower or higher than others in the general population varied within and between the groups.

*Other populations, I think it's the same?* FG1, woman, London.

*It's higher in Asian.*

*But tell me this, I have never heard of any Asian person with ovarian cancer.* FG4, women, Luton.

Most participants had not heard of genetic testing for cancer risk and those that had did not know about the specific *BRCA1/BRCA2* genes. After participants had been informed about genetic testing within the group, it became apparent that some had difficulty understanding that (1) the test would provide information about a person's risk rather than a cancer diagnosis and (2) being at risk did not mean that they would definitely develop cancer.

### Attitudes towards genetic testing and finding out about OC risk

Based on our presentation of genetic risk and genetic testing for OC, in discussion most participants initially expressed positive views. They felt they would benefit from knowing if they were at increased risk because they could take steps to manage their individual risk.

*So you are aware of it, and you know how to prevent it, getting information, what are the risks, and how to do your daily activity, your daily lifestyle, maybe that can change…* FG2, woman, London.

The majority of women indicated that they would accept genetic testing, and several men said that they would encourage female family members to have testing if it were offered, although they acknowledged that ultimately it would be the individual's decision. Many participants remained positive about genetic testing even after being told that risk information could potentially be less accurate for ethnic minorities. Participants said there were no cultural or religious prohibitions on genetic testing for cancer risk and these aspects of the discussion prompted some Muslim participants to speak of the positive influence that religion has on maintaining good health. In one group, participants referred specifically to the Imam (Muslim religious scholar), and his role in providing guidance to the community on health practices.

*Your religion wants you to look after yourself.* FG1, woman, London.

The main concerns voiced about genetic testing for OC risk related to experiencing worry between the blood test and receiving the result, and fear regarding the psychological impact of a high-risk result and what such a result would lead to if received.

*Until you know the outcome, your brain will be working overtime.* FG4, woman, Luton.

Participants were not fearful of providing blood for the genetic test and some indicated that while they were positive about genetic testing, others might be wary since they may be unfamiliar with OC, genetic testing and fearful of cancer per se.

*Negative side could be some people, maybe my mother, wouldn't wanna go to that test, maybe she would be scared, even if she doesn't have cancer…* FG2, woman, London.

Other negative aspects, such as the cost to the government or concerns that the test or 'diagnosis' could be wrong or inconclusive, were infrequently discussed.

## Attitudes towards risk-stratified management

Participants endorsed the risk-stratified management approach and accepted the information that there would be clear options for women at each level of risk. No concerns were expressed about receiving different treatment based on level of risk.

*…you will be able to find out what you have got and according to that you can prevent your, you know, things as well, if you want to go for like a screening or for a minor surgery or whatever it is, it's good to know what you have.* FG1, woman, London.

The screening component of risk-stratified management was generally endorsed, and participants felt that within their SA communities participation in current UK cancer screening programmes was gaining traction.

*Well at least it's something for your health, good health.* FG3, woman, London.

*It's best to take a test, best to take a test not to get to that stage, isn't it?* FG7, man, Luton.

However, participants talked about there still being some within the SA community who do not accept cancer screening per se. A few participants, mostly men, suggested that some women may not see the need to attend screening in the absence of symptoms. Reluctance to attend screening was also attributed to fear and issues of body privacy and shyness. Women owned that exposing their body to 'someone else' could be distressing, irrespective of whether the person was a healthcare professional.

*What she's saying is that first, Muslims were a bit scared and they wouldn't get checks done. They thought that maybe someone else would see them…* FG4, woman, Luton.

Attitudes towards risk-reducing surgery were mixed. While surgery was acceptable if it was deemed necessary by doctors, men and women said that women would not want surgery unless they had already had children and were of an older age.

*It depends on age because, any lady who is 50 years up, that time is high risk, she needs to remove that, but 25 years, 30 years, any lady, she has still option for children, so she can't do that thing.* FG2, woman, London.

Some men did not agree with risk-reducing surgery and believed that women would not accept it until cancer was diagnosed especially since 'high risk' did not mean the woman would definitely develop OC.

*Until such a time that a person is diagnosed with cancer, I don't think they will have their ovaries removed.* FG5, man, Luton.

## Family, culture and religion

The majority of women anticipated that their husbands and immediate families would support them if they had genetic testing for OC risk and most of the men also said that they would support their female family members if they wanted to be tested. Several women said they would discuss this beforehand with their husbands and families and/or would share the result.

*And then obviously going back and discussing it with family what's come up as well.* FG3, woman, London.

Both men and women felt that some women would keep genetic testing and their result secret or 'confidential' from family and friends:

*Some people may not want to share it with their family, want to keep it to themselves.* FG2, woman, London.

Some women, who found the PROMISE programme personally acceptable, had concerns that younger women found to be at high risk for OC could have difficulties finding a husband due to the pronatalist (promotion of human reproduction) nature of SA culture that favours the healthiest women. Others expressed concern that identifying someone as being at high risk or deciding to remove the ovaries could jeopardise existing marriages.

*…if you get to our age then we would say yes. But for people like my daughter I would advise her not to. Because maybe the husband would leave the wife.* FG2, woman, London.

Some men indicated that they would not ask a potential partner about their cancer risk and that this would not influence their marital choice. Others spoke of the cultural importance of marriage and a woman's ability to bear children where removal of the ovaries would be a serious issue.

A few participants expressed the view that illnesses such as cancer are predestined and come from God. However, they did not suggest that this meant they would do nothing to prevent illness; instead religion was referred to as a coping resource.

*…if I find out that I have got this problem; there will going to be ovarian cancer. I would thank God for giving me time to do what I want.* FG1, woman, London.

## Accessing services

Participants were eager for more information to be provided and for OC awareness campaigns within their communities.

*They don't even know where it is…Where is the problem? How does it happen? It is very important that we give this information first.* FG4, woman, Luton.

A variety of methods to reach SA women with key OC prevention, earlier diagnosis and risk management messages were discussed including community-based group sessions and campaigns in the local media targeted at those for whom English is not a first language. Several

participants suggested that as English was not their first language or because they could not read English, they would likely encounter difficulties in accessing genetic and screening services.

Both men and women indicated that, depending on the individual and situation, SA women would need or prefer to see a female healthcare professional, particularly if a physical examination was needed.

*Mostly, the issues are about female doctors checking female things and male doctors checking males.* FG5, man, Luton.

## DISCUSSION

In this qualitative study, we identified a worrying lack of awareness of OC among SA women and men. Others have reported low awareness of cancer risk factors and symptoms among UK ethnic minority groups,[25–28] although research has also shown a lack of OC symptom awareness among the UK general population.[28] Irrespective of whether risk-stratified OC management is offered to the public, improving awareness of OC among SAs is a health priority.

Some participants found it difficult to understand the brief presentation provided in the focus groups, it sometimes took several explanations to ensure that participants understood that genetic testing provides information on cancer risk rather than a diagnosis and that high risk did not mean that a person would definitely get cancer. As previously reported[11 12] participants tended to dichotomise risk as either high or low, with little discussion of intermediate risk. This underlines the need to develop optimal methods of conveying both the concept of OC risk and its meaning to the individual.[26 27] The challenges of communicating risk estimates to the lay public are well documented and are particularly challenging[29–31] when information materials need to be acceptable to diverse populations.

Our main finding that attitudes towards genetic testing for OC risk and stratified management were mostly positive is consistent with other studies,[32] but our study identified important cultural nuances. Participants maintained that personal genetic testing would not be viewed negatively from a religious standpoint and while a few referred to illness and death as predestined or from God, they indicated that it was still necessary to take action to maintain good health and, as in other research,[33] religion was referred to as a coping strategy. Cancer fatalism was infrequently identified, but in this and other research,[13] it was clear that the process of genetic testing and the receipt of a high-risk result was anticipated by several participants to create heightened anxiety. Genetic testing for OC risk and stratified management may not be acceptable to all SA women, in particular younger women. While the majority of women in this study indicated that they would accept a genetic test for OC risk if offered, many were already married and had children. Echoing research with UK Pakistanis about prenatal genetic testing[34 35] and UK

SA women with breast cancer,[36] some participants were concerned that illness or being identified as at high risk of OC could damage younger women's marriage prospects or cause marital problems. Participants acknowledged that not all SA women would discuss genetic testing or results with their family. Reluctance to discuss illness with family and friends due to taboo and perceived stigma was identified in the current study as well as in several other studies with SA participants,[36 37] and could act as a barrier to the uptake of genetic testing.

Participants accepted the idea of stratified risk management, that is, that there would be different management options for women with different levels of risk. With regard to the screening element of risk management, uptake of breast and cervical screening in the UK is lower among SA than white women.[38] While the situation is slowly improving among SA women generally, change has not been significant for Muslim SA women.[38] Interestingly, in our study several groups discussed a positive cultural change in attitudes towards cancer screening and the majority reported attendance at breast and/or cervical screening. In line with previous research,[39–41] participants argued that lack of awareness, embarrassment and shyness were barriers to attending screening. While OC screening with blood tests and ultrasound scans was seen as acceptable by most participants, the study did not explore the acceptability of transvaginal ultrasound (the most commonly used scan to help detect OC), as this was beyond the scope of the study. However, as reported elsewhere,[14 33] participants did have a preference for consultations with a gender-matched healthcare professional, particularly if a physical examination was needed.

Risk-reducing oophorectomy was seen as a particular dilemma, principally due to the importance placed on women's ability to bear children. Some male participants felt that it would be better to wait and see if a cancer develops, catch it at an early stage and then have surgery. It may be that these men were inappropriately applying to OC their knowledge of how other cancers present and develop and this needs further investigation. However, apprehension about this surgery is not unique to SAs.[42] Our study highlights the need for sufficient information and support to be offered to SA women considering predictive genetic testing, and particularly for those with increased risk who will need to make risk management decisions.

This is the first study to explore UK SAs' perspectives on population-based genetic testing and risk-stratified management for OC, and includes participants with various levels of English language who are often not included in research. However, the opinions expressed by participants with regard to genetic testing and the PROMISE programme were based on brief information which was new to all, and related issues such as insurance and ethics were not spontaneously discussed. Furthermore, the current study did not inform patients that an increased risk of OC due to a *BRCA* gene mutation also indicates an increased risk of breast cancer, as this was

beyond the scope of the study. The risk of breast cancer would further complicate decision-making as high-risk patients would need to consider increased surveillance for breast cancer or risk-reducing mastectomy. Also, both male focus groups were run by female facilitators; while participants did not express dissatisfaction with this, it may have influenced their responses.

## Conclusions

Population-based risk assessment and stratified management may be acceptable to many SA men and women in the UK. Attitudes towards cancer screening were positive; however, opinions on risk-reducing surgery were mixed. The study highlights a need for tailored OC awareness campaigns within SA communities. To be inclusive, genetic testing and aftercare services should accommodate non-English speakers, offer appointments with a gender-matched healthcare professional and offer patients support with their healthcare decisions.

**Author affiliations**
[1] Department of Women's Cancer, EGA UCL Institute for Women's Health, University College London, London, UK
[2] Health Psychology Research Unit, Royal Holloway, University of London, Egham, UK
[3] Institute for Health Research, University of Bedfordshire, Luton, UK
[4] Department of Clinical Genetics, University Hospital Southampton NHS Foundation Trust, Southampton, UK
[5] Department of Behavioural Science and Health, Institute of Epidemiology and Health Care, University College London, London, UK
[6] Department of Clinical Genetics, Great Ormond Street Hospital, London, UK

**Acknowledgements**  This work was carried out at UCLH/UCL within the Cancer Theme of the NIHR UCLH/UCL Comprehensive Biomedical Research Centre supported by the UK Department of Health for the PROMISE study team. We thank all the women and men who took part in the focus groups and all the centres and community group leaders who helped to recruit participants and distribute posters for the study.

**Contributors**  Authors KEJH, NA, LSMF, LS, JW, SCS and AL contributed to the design of the study including refinement of the discussion guide. AL was overall responsible for the delivery of the project. NA and KEJH recruited participants and conducted the seven focus group discussions. KEJH analysed the data and NA and AL checked the data for any divergent cases. Independent researcher SG performed coding checks on a proportion of the data. Authors KEJH, NA, LSMF, LS, JW, SCS, SG and AL contributed to interpretation of the results. KEJH drafted the manuscript and authors KEJH, NA, LSMF, LS, JW, SCS, SG and AL critically reviewed and approved the manuscript.

**Funding**  This work was supported by The Eve Appeal (509050) and Cancer Research UK (C1005/A12677).

**Disclaimer**  The funders had no role in the study design; collection, management, analysis or interpretation of data; writing of the report or the decision to submit the report for publication.

**Competing interests**  None declared.

**Patient consent**  Not required.

**Ethics approval**  Approval to conduct the study was granted by the UCL Research Ethics Committee (project ID: 8053/003).

**Provenance and peer review**  Not commissioned; externally peer reviewed.

**Data sharing statement**  Anonymised qualitative data are available on request.

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
