## [Reviewer comments · BMJ Open]

ARTICLE DETAILS

TITLE (PROVISIONAL)	Attitudes towards a programme of risk assessment and stratified management for ovarian cancer: A focus group study of UK South Asians' perspectives
AUTHORS	Hann, Katie; Ali, Nasreen; Gessler, Sue; Fraser, Lindsay; Side, Lucy; Waller, Jo; Sanderson, Saskia; Lanceley, Anne

VERSION 1 – REVIEW

REVIEWER	Robin Crawford Cambridge University Hospitals NHS Foundation Trust
REVIEW RETURNED	15-Feb-2018

GENERAL COMMENTS	This is a nice piece of qualitative research on a complex topic. The UCL group have already looked at population screening in the Jewish community who carry a higher incidence of BRCA gene mutation. This piece of work does seem to be quite narrow in addressing only south Asians rather than a wider group of ethnic minorities. The work also highlights the significant ignorance about ovarian cancer, its development and its outcome. Moving onto the complexity of a triage depending on relative risk of the well person is very hard to grasp and understand in a focus group setting - in the 75 minute meeting there is a lot to discuss and consider especially in relation to the intervention of risk reducing surgery. The limitation of only discussing ovarian cancer and not considering the intervention for breast cancer are interesting - it would have been a useful comparison to discuss the breast approach - In OC screening does not work (Rosenthal 2012) and in breast cancer screening is a viable alternative to risk reducing surgery. It may be useful to consider a comparator group such as black British and also the East European new immigrants.
---

REVIEWER	Chew Kah Teik Universiti Kebangsaan Malaysia Medical Center
REVIEW RETURNED	04-Mar-2018

GENERAL COMMENTS	Congratulations to Hann and team for their work in such an excellent study. Worldwide, the voice of minority ethnics often been neglected in health care policy. The authors critically discuss this issue in this interesting manuscript.
--

REVIEWER	Gillian Hanley University of British Columbia, Vancouver, Canada
REVIEW RETURNED	21-Mar-2018

GENERAL COMMENTS	This is an interesting and well written study on the attitudes and knowledge of South Asians in the UK towards ovarian cancer,
--

	ovarian cancer screening, and ovarian cancer prevention. 1) While not entirely relevant to this article, as it never really gets into the risk prediction in any meaningful way (see comment 4 below), it's not clear to me what the addition of the non-genetic information truly adds. Given how drastically a BRCA mutation increases risk for both breast and ovarian cancer, it seems the focus should be on population-based BRCA mutation at this time. If a cost-effective screening strategy is eventually identified, then considering more nuanced risk prediction would be more relevant, but at this point, we really want to identify BRCA mutations because risk is high enough to warrant interventions. There also isn't an issue with over-investigation of those at lowest risk if you focus on BRCA mutations. 2) The abstract does not currently provide a compelling rationale for studying this in the South Asian population. Perhaps introducing that it has been studied among white people in the UK and now the awareness and attitudes of the South Asian community are being assessed would be helpful. 3) Could you provide some rationale for the inclusion of the two focus groups with men? It's not immediately apparent why they would be included in a study on ovarian cancer. It becomes clear as you read the results, but it would be nice to have a sentence or two on your reasoning. 4) Perhaps placing these results in a broader context might be helpful. The introduction seems oddly specific to PROMISE, which is not well elucidated in the article, and actually doesn't seem that integral to the ultimate findings (see comment 1 above) which seems to be more commentary on SA views around ovarian cancer, ovarian cancer screening and ovarian cancer prevention in general. I think the introduction needs to be rewritten to focus more on what is actually being accomplished in this article and less on PROMISE. I understand that is ultimately what your research group is interested in, but it actually has very little relevance to the article, and leaves the reader confused because they are expecting something very different.
--	---

REVIEWER	Tomasz Kluz District Hospital No.1 Rzeszow Poland , Dept Ob/Gyn
REVIEW RETURNED	31-Mar-2018

GENERAL COMMENTS	I have no reservations
------------------------

VERSION 1 – AUTHOR RESPONSE

Reviewer: 1

Reviewer Name: Robin Crawford

Institution and Country: Cambridge University Hospitals NHS Foundation Trust

Please state any competing interests: None declared

Please leave your comments for the authors below

This is a nice piece of qualitative research on a complex topic. The UCL group have already looked at population screening in the Jewish community who carry a higher incidence of BRCA gene mutation.

This piece of work does seem to be quite narrow in addressing only south Asians rather than a wider group of ethnic minorities. The work also highlights the significant ignorance about ovarian cancer, its development and its outcome. Moving onto the complexity of a triage depending on relative risk of the well person is very hard to grasp and understand in a focus group setting - in the 75 minute meeting there is a lot to discuss and consider especially in relation to the intervention of risk reducing surgery.

The limitation of only discussing ovarian cancer and not considering the intervention for breast cancer are interesting - it would have been a useful comparison to discuss the breast approach- In OC screening does not work (Rosenthal 2012) and in breast cancer screening is a viable alternative to risk reducing surgery. It may be useful to consider a comparator group such as black british and also the east European new immigrants.

Thank you for your helpful comments.

Only ovarian cancer was discussed with participants because the main focus of the PROMISE programme that this study is part of is to improve earlier detection specifically for ovarian cancer through risk prediction and screening. We decided against discussing breast cancer risk in addition to ovarian cancer risk as we believe this would likely have overburdened participants with information and proved difficult to cover within the 1 ½ hours allowed for the focus groups.

We identified a South Asian sub-population of Black and Minority Ethnic people living in the UK for our study as South Asians are the largest non-European ethnic minority group in the UK. We did this to increase the likelihood of our study delivering meaningful results. To this end we also purposively sought the opinions of women and men from a range of South Asian communities: Indian, Pakistani and Bangladeshi.

It would indeed be interesting and important to consider comparator groups in a future study and address the broader black and minority ethnic community living in the UK.

Reviewer: 2

Reviewer Name: Chew Kah Teik

Institution and Country: Universiti Kebangsaan Malaysia Medical Center

Please state any competing interests: No issue of competing interests.

Please leave your comments for the authors below

Congratulation to Hann and team for their work in such an excellent study. Worldwide, the voice of minority ethnics often been neglected in health care policy. The authors critically discussion this issue in this interesting manuscript.

Thank you for your positive review.

Reviewer: 3

Reviewer Name: Gillian Hanley

Institution and Country: University of British Columbia, Vancouver, Canada

Please state any competing interests: None declared

Please leave your comments for the authors below

This is an interesting and well written study on the attitudes and knowledge of South Asians in the UK towards ovarian cancer, ovarian cancer screening, and ovarian cancer prevention.

1) While not entirely relevant to this article, as it never really gets into the risk prediction in any meaningful way (see comment 4 below), it's not clear to me what the addition of the non-genetic information truly adds. Given how drastically a BRCA mutation increases risk for both breast and ovarian cancer, it seems the focus should be on population-based BRCA mutation at this time. If a cost-effective screening strategy is eventually identified, then considering more nuanced risk prediction would be more relevant, but at this point, we really want to identify BRCA mutations because risk is high enough to warrant interventions. There also isn't an issue with over-investigation of those at lowest risk if you focus on BRCA mutations.

Thank you for carefully considering our work.

Our qualitative study is part of a programme of research called PROMISE which has developed an algorithm for predicting ovarian cancer risk and has investigated possible novel biomarkers to include in this. A feasibility trial investigating uptake and acceptability of the resulting ovarian cancer risk prediction and stratified management is underway. We understand that the presence of a BRCA mutation is the largest risk predictor for both breast and ovarian cancer. However the aim of the PROMISE study is to factor in other risk predictors to categorise woman's risk for ovarian cancer into low, intermediate and high. Our study investigates South Asian women and men's attitudes to the PROMISE programme of risk prediction and risk stratified management for ovarian cancer. We consider the description of PROMISE to be important and sufficient in our introduction and that it would be inappropriate to rewrite our introduction.

2) The abstract does not currently provide a compelling rationale for studying this in the South Asian population. Perhaps introducing that it has been studied among white people in the UK and now the awareness and attitudes of the South Asian community are being assessed would be helpful.

We provide a rationale for seeking the views of a more diverse UK population on page 4/5 of our manuscript. The awareness and attitudes of the South Asian community was investigated because this is one of the largest ethnic minority groups in the UK and their opinions are often under-represented in research. Our rationale for investigating this topic with the South Asian community has been added to the abstract.

3) Could you provide some rationale for the inclusion of the two focus groups with men? It's not immediately apparent why they would be included in a study on ovarian cancer. It becomes clear as you read the results, but it would be nice to have a sentence or two on your reasoning.

Men were included in this study because it is known that healthcare decisions are often discussed within families and we were interested in what men knew already on the topic and what their attitudes to the programme might be as this could also influence women's attitudes and healthcare decisions. Our rationale, with supporting evidence has been added to the introduction on page 5.

4) Perhaps placing these results in a broader context might be helpful. The introduction seems oddly specific to PROMISE, which is not well elucidated in the article, and actually doesn't seem that integral to the ultimate findings (see comment 1 above) which seems to be more commentary on SA views around ovarian cancer, ovarian cancer screening and ovarian cancer prevention in general. I think the introduction needs to be rewritten to focus more on what is actually being accomplished in this article and less on PROMISE. I understand that is ultimately what your research group is interested in, but it actually has very little relevance to the article, and leaves the reader confused because they are expecting something very different.

We thank the reviewer for this comment but for reasons explained above we do not feel able to make fundamental changes to the introduction. To do so would invalidate our study and its results.

Reviewer: 4

Reviewer Name: Tomasz Kluz

Institution and Country: District Hospital No.1 Rzeszow Poland, Dept Ob/Gyn

Please state any competing interests: none declared

Please leave your comments for the authors below

I have no reservations

Thank you for your positive review.

VERSION 2 – REVIEW

REVIEWER	Gillian Hanley University of British Columbia, Canada
REVIEW RETURNED	16-May-2018
GENERAL COMMENTS	The revisions are sufficient.